# Factors affecting integration of an early warning system for antimalarial drug resistance within a routine surveillance system in a pre-elimination setting in Sub-Saharan Africa

Frank M. Kagoro[1,2,3,4,5], Elizabeth Allen[1,3,4], Jaishree Raman[6,7,8], Aaron Mabuza[1,3], Ray Magagula[9], Gerdalize Kok[9], Gillian Malatje[9], Philippe J. Guerin[3,4,5], Mehul Dhorda[2,3,4,5], Richard J. Maude[2,5,10,11], Karen I. Barnes[1,3,5]*

1 Collaborating Centre for Optimising Antimalarial Therapy (CCOAT), Division of Clinical Pharmacology, Department of Medicine, University of Cape Town (UCT), Cape Town, South Africa, 2 Mahidol Oxford Tropical Medicine Research Unit (MORU), Faculty of Tropical Medicine, Mahidol University, Bangkok, Thailand, 3 WorldWide Antimalarial Resistance Network (WWARN), South-East African Regional Hub, UCT, Cape Town, South Africa, 4 Infectious Diseases Data Observatory (IDDO), Oxford, U.K. 5 Centre for Tropical Medicine and Global Health, Nuffield Department of Medicine, University of Oxford, Oxford, United Kingdom, 6 Centre for Emerging Zoonotic and Parasitic Diseases, National Institute for Communicable Disease (NICD), Sandringham, Gauteng, South Africa, 7 Wits Research Institute for Malaria, Faculty of Health Sciences, University of Witwatersrand, Johannesburg, South Africa, 8 UP Institute for Sustainable Malaria Control, Faculty of Health Sciences, University of Pretoria, Pretoria, South Africa, 9 Mpumalanga Provincial Malaria Elimination Programme, Mpumalanga, South Africa, 10 School of Public Health, Li Ka Shing Faculty of Medicine, University of Hong Kong, Pok Fu Lam, Hong Kong, 11 The Open University, Milton Keynes, U.K.

* karen.barnes@uct.ac.za

## Abstract

To address the current threat of antimalarial resistance, countries need innovative solutions for timely and informed decision-making. Integrating molecular surveillance for drug-resistant malaria into routine malaria surveillance in pre-elimination contexts offers a potential early warning mechanism for further investigation and response. However, there is limited evidence on what influences the performance of such a system in resource-limited settings. From March 2018 to February 2020, a sequential mixed-methods study was conducted in primary healthcare facilities in a South African pre-elimination setting to explore factors influencing the flow, quality and linkage of malaria case notification and molecular resistance marker data. Using a process-oriented framework, we undertook monthly and quarterly data linkage and consistency analyses at different levels of the health system, as well as a survey, focus group discussions and interviews to identify potential barriers to, and enhancers of, the roll-out and uptake of this integrated information system. Over two years, 4,787 confirmed malaria cases were notified from 42 primary healthcare facilities in the Nkomazi sub-district, Mpumalanga, South Africa. Of the notified cases, 78.5% (n = 3,758) were investigated, and 55.1% (n = 2,636) were successfully linked to their *Plasmodium falciparum* molecular resistance marker profiles. Five

**Data availability statement:** All qualitative relevant data are within the paper and its Supporting Information files. The quantitative datasets generated and/or analysed during the study are publicly available from the WWARN Tracking Resistance website (https://www.wwarn.org/tracking-resistance/artemisinin-molecular-surveyor).

**Funding:** The Smart Surveillance for Malaria Elimination Pilot in Mpumalanga, South Africa, was co-funded by the South African MRC and the Worldwide Antimalarial Resistance Network (WWARN). WWARN is funded by the Bill and Melinda Gates Foundation and the ExxonMobil Foundation. This research was also, in part, funded by the Wellcome Trust [Grant Number 220211]. For the purpose of open access, the author has applied a CC BY public copyright licence to any Author Accepted Manuscript version arising from this submission. Funders did not have any influence in the design of the study, data collection, analysis, interpretation of data and or writing the manuscript.

**Competing interests:** The authors have declared that no competing interests exist.

tangible processes—malaria case detection and notification, sample collection, case investigation, analysis and reporting—were identified within the process-oriented logic model. Workload, training, ease of use, supervision, leadership, and resources were recognized as cross-cutting influencers affecting the program's performance. Approaching malaria elimination, linking molecular markers of antimalarial resistance to routine malaria surveillance is feasible. However, cross-cutting barriers inherent in the healthcare system can influence its success in a resource-limited setting.

## 1. Introduction

Sub-Saharan Africa bears the largest burden of malaria, with 94% and 95% of the global cases and deaths, respectively [1]. Malaria-endemic countries in sub-Saharan Africa are now also facing the threat of antimalarial drug resistance. Malaria parasites with reduced susceptibility to artemisinin-derivatives are emerging and rapidly spreading, threatening the continent's control and elimination goals, heightening the need for novel tools and strategies to effectively tackle this threat [2–8]. Such innovations need to be timely, relevant and tailored to existing health systems, particularly in resource-limited settings.

In its strategy to respond to antimalarial drug resistance in Africa, the World Health Organization (WHO) emphasised the need for robust and agile surveillance systems capable of promptly detecting and responding to antimalarial drug resistance and recommends that this surveillance be integrated into routine malaria surveillance systems [9,10]. This integration is particularly important in pre-elimination areas, where the risk of drug-resistant parasites emerging may be heightened by higher drug pressure and lack of partial immunity [11]. In these settings, the WHO also recommends monitoring the prevalence of molecular markers of antimalarial drug resistance, as an early warning system to identify resistance signals and areas requiring further investigation, including therapeutic efficacy studies (TES). Despite TES being the cornerstone of antimalarial drug efficacy monitoring, they are not conducted regularly and do not provide a timely spatial representation of the distribution of antimalarial resistance. Previous evidence has shown that integrating antimalarial molecular markers using a routine malaria notification system is feasible in sub-Saharan African settings [12]. However, little is known about the barriers and enhancers to integrating molecular markers of resistance within the routine malaria surveillance system. Understanding such factors is needed by countries adopting or expanding molecular surveillance, for which recent laboratory investments have been extensive [13,14].

In 2018, the South African National Malaria Programme (NMP) piloted a novel technology to consolidate and enhance malaria surveillance activities and treatment approaches fundamental to achieving malaria elimination. Through the Smart Surveillance for Malaria Elimination (SS4ME) initiative, antimalarial drug resistance molecular marker data were linked to malaria case notifications in near real-time in Nkomazi, Mpumalanga, a pre-elimination setting [15]. SS4ME included the collection of malaria rapid diagnostic tests (mRDTs) and, wherever possible, dried blood spots (DBS) on filter papers for assaying molecular markers of antimalarial resistance that can be linked by unique barcodes to individual case notifications.

Developing and evaluating the integration of molecular markers of antimalarial resistance into routine malaria surveillance required an assessment of the surveillance system's performance, both independently and together with other routine notification components, making for a complex intervention [16]. As per national malaria treatment guidelines, all suspected malaria cases presenting to health facilities should be confirmed by a mRDT or microscopy (passive case detection) before treatment is administered [17]. Additionally, in pre-elimination areas of South Africa, the NMP screens high-risk groups, such as migrant and mobile populations (proactive case detection), and households surrounding the residence of index cases (reactive case detection) [6,7]. Therefore, integration of molecular resistance markers into routine malaria surveillance systems involved adopting several technologies, activities, and processes into the existing malaria notification system, with regular monitoring and adaptations. For SS4ME, implementation focused on exploring and guiding how the roll-out, adoption and utilisation of new and existing technologies would advance malaria elimination goals. The SS4ME rollout would involve optimising and linking existing technologies with key functions situated at health facilities, such as malaria testing, treating, referral, and notification provincial level, collation of malaria notification metrics, and academic institutions assaying for molectular markers of antimalarial resistance. This laid a foundation for benchmarking how SS4ME was received and understood by potential key stakeholders. As with other implementation research, this roll-out lacked a mechanism to explore the interaction between the different determinants of success and the users or beneficiaries. Therefore, this approach needed to be revised and expanded to identify internal, external and interactive factors that could affect the implementation and impact of the intervention. Through the step-by-step depiction of how a program operates and how it leads to the desired outcomes, process-oriented logic models have been used in programme planning and evaluation to understand its key elements [18].

'Making Data Mapworthy' was a quantitative sub-study linked to the SS4ME pilot that evaluated the feasibility of integrating molecular resistance markers into the routine malaria surveillance system using coverage, accuracy and linkage of malaria cases in near real-time [12]. However, these metrics alone could not fully examine the flow of data, or users' practices and perceptions. Here, factors influencing the flow, quality and linkage of malaria notification data and associated molecular resistance markers from different reporting levels are investigated to inform the enhancement and sustainability of this integrated early warning system for antimalarial resistance.

## 2. Materials and methods

This SS4ME sub-study used an iterative sequential mixed-methods design and included 1) monthly and quarterly quantitative descriptive analyses, 2) a healthcare facility staff survey, and 3) focus group discussions (FGDs) and in-depth interviews (IDIs) with healthcare facility staff involved in malaria case management and NMP staff in Nkomazi sub-district, a pre-elimination area in Mpumalanga province, South Africa.

Individuals identified through proactive, reactive, or passive case detection were screened for malaria using a falciparum-specific histidine-rich protein 2 (HRP2)-based mRDT (First Response™ Malaria Ag *P. falciparum* HRP2 Detection Rapid Card Test, Premier Medical Corporation Ltd, India) selected according to the Mpumalanga Malaria Programme tender process. For molecular surveillance, DBS filter papers were collected from all patients with positive mRDTs. An additional 10% of the negative mRDTs were collected and sent to the National Institute for Communicable Diseases (NICD) for quality assurance. Both symptomatic and asymptomatic positive cases were treated following national treatment guidelines with the WHO-recommended 3-day artemether-lumefantrine regimen, used in the area since 2007. Demographic and case data were collected through the Notifiable Medical Condition form or app and verified for quality within 24 hours at the sub-district NMP office. Case investigators conducted household visits within 24–72 hours of notification, recording Global Positioning System (GPS) coordinates and assessing malaria risk factors. All notifications were quality-checked and electronically captured into the routine District Health Information System II (DHIS2) at the sub-district NMP office [17].

Three levels of notification data were examined: healthcare facilities, the sub-district NMP office serving as a data collection/ capturing centre and provincially through DHIS2 submissions. All SS4ME-participating healthcare facilities were enrolled for malaria case data analysis at monthly and quarterly intervals, to gauge coverage and consistency as performance metrics. The facilities were then categorised as low- or high-performing based on the overall data linkage, and two from each level were purposively selected for enhanced data quality assessment.

A paper-based survey tool (S1 Tool) was collaboratively developed by the study team and was revised by the Mpumalanga Malaria Programme and approved by the University of Cape Town Health Research Committee. S2 Tool and S3 Tool (Informed consent and FGD and IDI guide) were similarly developed, with input from subject matter experts in the team to ensure relevance and appropriateness. All tools were piloted and field-validated in collaboration with the malaria programme team, who tested them, provided feedback, and ensured their alignment with the study objectives before formal deployment.

S1 Tool (staff survey tool) was administered between 1st March and 30th June 2020 to primary healthcare facility staff treating malaria patients to evaluate their practice, perception, and experience of the integration of molecular resistance markers into routine malaria surveillance system activities. For FGDs and IDIs, various cadres of staff involved in malaria case management at healthcare facilities and in the NMP were invited to participate by email or phone call. At least one staff member performing any malaria-related activities was invited per healthcare facility. After obtaining verbal and written consent using S2 Tools (b) and (c), FGDs and IDIs were conducted in English from 3rd to 6th June 2020 using pre-prepared FGD and KII guides (S2 Tool and S3) to maintain consistency and quality. The audio recordings were securely stored on the password-controlled study computer and later transcribed. Two study investigators listened to audio recordings with reference to the transcribed scripts and resolved any interpretation conflicts or transcription errors by consensus.

This study was approved by the Human Research Ethics Committee of the Faculty of Health Sciences at the University of Cape Town (HREC REF Number: 698/2019 and 038/2020, including for data obtained in the SS4ME study (HREC REF Number: 519/2017). SS4ME was also endorsed by the South African Department of Health, the NICD and the Mpumalanga Malaria Programme. All partners were notified, and their staff were informed of the study, including NMP staff, information officers, clinicians and data clerks.

The descriptive analysis focused on staff survey data and monthly and quarterly DHIS2 data evaluations. The latter summarised notified and investigated cases, mRDT/ DBS samples analysed, their linkage, as well as spatiotemporal trends in molecular resistance markers and usability assessments. All quantitative data analyses were conducted using R programming language (versions 3.6 and 4.0). Further quantitative methods used include spatial temporal, trend and usability analysis which have been explained elsewhere [12]. For consistency, quarterly aggregates of notified cases and investigated cases from each healthcare facility sampled were compared at three levels (healthcare facility, sub-district data capture centre and provincial DHIS2 records) and the median difference was computed. Consistency was defined as an equal number of cases being reported at different levels, with a difference of +/- 5 cases allowed to account for delayed reporting. Since a consistency benchmark for integrating molecular resistance data into malaria case data had not been described before, this was established by adapting the internal consistency benchmark proposed by the WHO Data Quality Assurance guideline [19]. Data were then explored quantitatively and qualitatively to identify possible causes of inconsistency.

For the qualitative data, each audio recording was transcribed and imported to NVivo 12 before being coded deductively, based on the interview guides and the process-oriented logic model (S1 Table), and inductively from other observations. The first coding cycle assigned labels to text excerpts in the transcripts as lowest level 'nodes'. These were then explored for repeated ideas and patterns, which were grouped, organised and categorised as higher-level themes, based on the interview guide sections and new concepts emerging. The prevailing themes and their theories were extracted and further analysed through the framework matrix, and the analytical outputs were evaluated across the range of participants and stakeholder groups.

 

As a complex intervention, the study used a process-oriented logic model to link the overall qualitative and quantitative data to determine the enablers and barriers that would explain the changing trends. To do so, results were further categorised into processes and logically analysed to identify inputs (e.g., resources, technology components) and activities (e.g., implementation steps, training), intended and unintended outputs (e.g., data linkage, use practices), and outcomes (e.g., integration success, surveillance improvements). A matrix was constructed to explore the level of influence and performance for each key feature identified from the process logic model and is further presented in the results section.

## 3. Results

A total of 4,787 malaria cases were notified in Nkomazi, Mpumalanga with 78.5% (n = 3,758) investigated by the NMP (Fig 1) from March 2018 to February 2020. Of the notified cases, 55.1% (n = 2,636) were linked to their *Plasmodium falciparum* molecular resistance marker profiles, with 85% (n = 2,240) of the linked cases mapped to healthcare facility, ward and locality levels. No validated or associated artemisinin-resistant Kelch-13 mutations were identified among the 2,385 PCR-positive samples, while nearly all 2,812 samples analysed for lumefantrine susceptibility carried the wild-type *mdr*1 86ASN and *crt* 76LYS alleles, potentially linked to reduced lumefantrine susceptibility. Concurrently, the spatial accuracy of malaria case coordinates improved, with the average nearest neighbour distance decreasing from 330 km in the second quarter to 35 km by the fifth quarter. Further quantitative results of spatial temporal, trends, usability and molecular markers are reported elsewhere [12].

A total of 46 healthcare workers participated in five separate FGDs, with group sizes of 8–12 participants. Of the participants, 32 were nurses (including professional nurses, occupational nurses, and assistant nurses), and 14 were environmental health officers (who worked as health promoters or malaria case investigators), data clerks, data capturers and surveillance officers. The majority of the participants were women (n = 31). Four IDIs were conducted with staff at the supervisory and decision-making levels performing infection control coordination, district and provincial NMP management, primary healthcare management and malaria surveillance supervision.

The 50 participants involved in the FGDs and IDIs had between 1 and 31 years of experience working with the NMP and were involved in the surveillance data reporting chain. Their responsibilities included field and clinical diagnostics, care of patients, case notification, case investigation, data capture, sample collection, labelling and transport, supervisory and reporting roles and decision-making. Several key processes emerged from discussions on the integration of

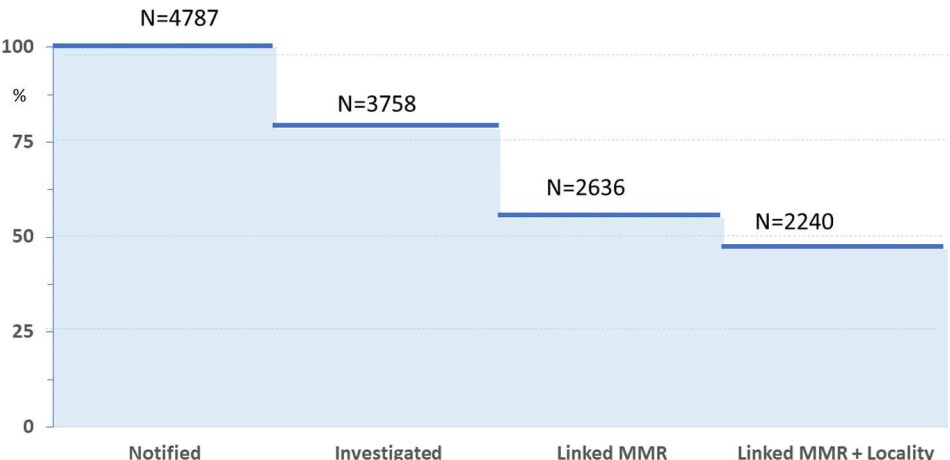

**Fig 1. Summary of cases notified, investigated, and linked to molecular markers of resistance (MMR) and locality from March 2018 to February 2020 in Nkomazi sub-district, Mpumalanga.**

molecular resistance markers into the routine malaria surveillance system, which included malaria case detection and notification, sample collection, packaging and transportation, case investigation, data capture, analysis and reporting.

A total of 64 nurses from 21 of the 42 healthcare facilities participated in the survey. Respondents were registered/professional nurses (67.1%), enrolled nurses (22%) and occupational nurses (6%). Two respondents did not mention their job titles. Five quarterly assessments were conducted for three healthcare facilities for the consistency evaluation. Due to logistical challenges, the team did not manage to enroll the fourth health facility.

In Fig 2 below, prevalent themes in the FGDs and IDIs included ease of use, perceived usefulness of surveillance, staff members' reluctance to adopt new activities, contradictions in best practice definitions, workload, and system support. These themes converged into sub-themes: work commitment, agency and ownership, challenging processes, compromise, staffing needs, training, leadership, and supervision.

Five key processes (S1 Table) emerged: malaria case detection and notification, sample collection, case investigation, data capture, analysis and reporting. These themes are described below using the process-oriented logic model outlined in S1 Table, linking them to processes, inputs, outputs, and outcomes.

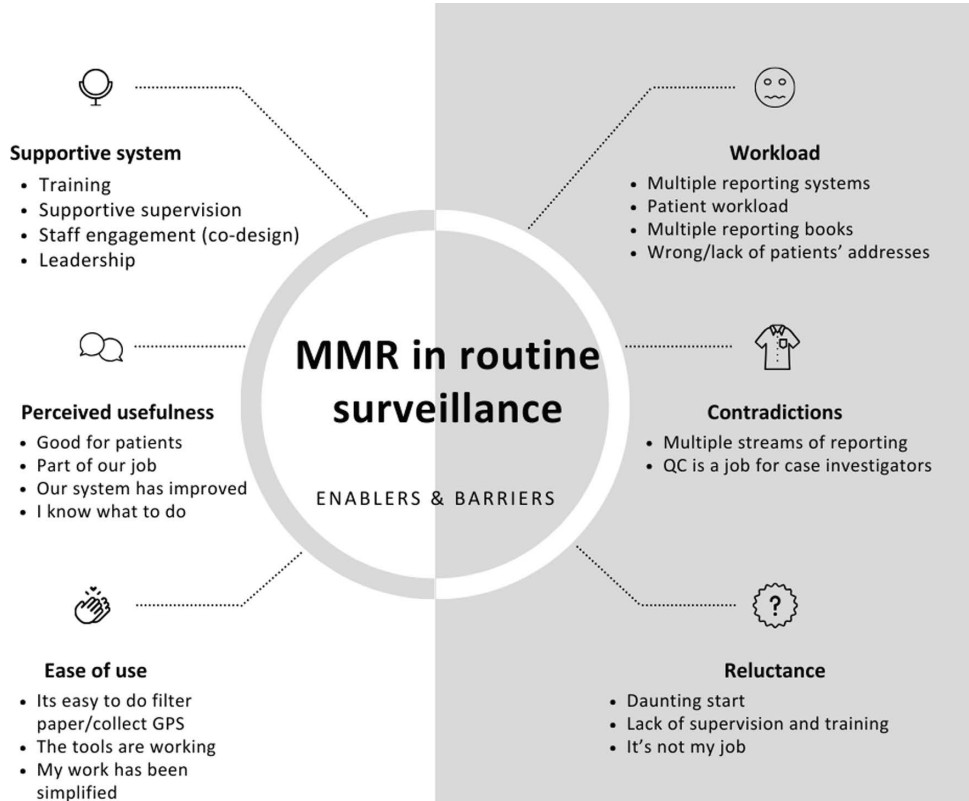

**Fig 2. The factors discovered during focus group discussions and in-depth interviews influencing the implementation of molecular resistance markers into the routine malaria surveillance system in the Nkomazi sub-district.** Several themes and sub-themes surfaced from these discussions. Initially, participants voiced reluctance and contradictions, leading to confusion over roles. This uncertainty spawned a challenging start and negatively impinged on the work. To counter these issues, more intensive training, stauncher leadership, and consistent supervision were recommended. In instances where confusion persisted, a supportive system was seen as beneficial to harmonise tasks and provide necessary training and quality control. This tactic successfully alleviated immediate challenges. Lastly, even though the integration process demanded an increased workload, healthcare staff who were willing to learn and perceived the benefits of this integration displayed a heightened commitment to their work. This may also indicate a need for increased staffing or a reorganised work system to provide adequate support.

### 4.1 Process I: Malaria case detection and notification

The change introduced during the integration of molecular surveillance for resistance markers into the routine malaria surveillance system was the inclusion of barcode stickers on the case notification forms during passive, proactive and reactive case detection. The pilot also developed activities to enhance the collection of all required malaria case details in the reporting notification form (e.g., contact information and household address/ directions). This was to support the linkage of demographic and location data to the molecular markers of resistance. This also resulted in additional workload, which was aggravated by the multiple notification reporting systems. As shown in S1 Fig, the Ministry of Health and NMP introduced two new malaria notification systems, leading to three malaria notification systems running concurrently during the study period, namely a) the Notifiable Medical Condition (NMC) book/ forms, b) the Malaria Connect mobile application and c) NMC mobile application.

The surveillance team supervisors mentioned that healthcare workers routinely used the Malaria Notification Book for notification, and only notified cases using other systems as an additional option. This was corroborated by healthcare workers themselves who reported using the notification book first and other notification methods depending on the workload, as quoted below:

*'We do the paper first, the notification book first, the remaining depends on their time or workload'.* **[Nurse, IDI06].**

However, in responding to the survey question on which systems are used frequently for notification, 61/64 participants reported using these interchangeably: Malaria Connect (36, 59%), NMC Notification Book (15, 25%) and NMC Mobile Phone Application (3, 5%). Of those who reported using more than one system, only one (2%) reported using all three, while five (8%) used Malaria Connect and NMC Notification Book only, and one (2%) used both Malaria Connect and NMC Phone App (Fig 3).

Participants reported that the parallel reporting systems introduced an additional burden to their daily work, particularly in the community health centres (CHC) operating 24-hours per day, 7 days a week, where a high number of patients present. For example, one of the nurses from a CHC in the FGDs said:

*'But the volume you can look at the CHCs [Community Health Centres] and eight-hour [clinics], there are more people coming in because day and night we are working.'* **[Nurse FGD03].**

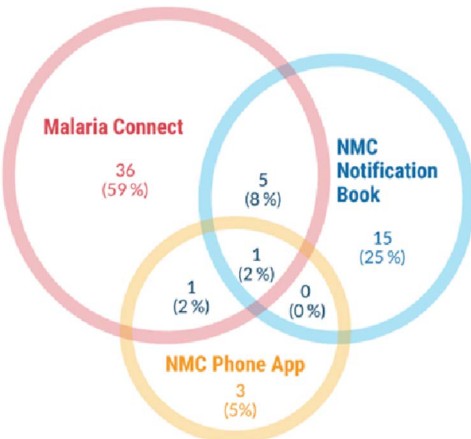

**Fig 3. Malaria notification systems used by 61 staff working in 21/42 healthcare facilities in Nkomazi, Mpumalanga, South Africa.** NMC: Notifiable Medical Condition.

## 4.2 Process II: sample collection

The project introduced the collection of DBS filter papers, with related SOPs at all participating healthcare facilities. Participants expressed reluctance about collecting DBS and capturing patient information on mRDTs. This called for increased programmatic support and training, and with time, the study observed improvement of barcoded mRDT samples received at the NICD laboratory from 19% in the first quarter to 85% in the final quarter, reflecting gradual acceptance and adaptability [12].

*'Basically, at first it was quite a daunting task because now [I] remember we had facilities that recently introduced the new NMC form and also Malaria Connect and all of that. So when we went to train the health facilities [about notification and DBS sample collection] they were resistant because they feel that it is extra work on their behalf...'* **[Environmental Health Officer, FGD01].**

Not all healthcare workers grasped the procedure for blood collection immediately. Due to difficulty in obtaining enough blood from the initial finger prick, they occasionally had to prick the patient's finger a second time. This caused some resistance among patients, according to the staff:

*'Yes, there is not enough [blood from one finger prick]. There is a little, but it is not enough. So, most of the time they [nurses] have to prick maybe twice, or three times sometimes.'* **[Case investigator, FGD01].**

However, this also encouraged the healthcare workers to understand the rationale behind filter paper sample collection, to be able to explain this to the patients who were hesitant:

*'These activities have helped us to know what we are doing and to help the patients; if they ask me now why we are taking another blood sample, it is for monitoring drug resistance and making sure you are cured. So, it's a very good thing'* **[Nurse, FGD04].**

After DBS sample collection, healthcare workers were asked to label and package samples, ready to be collected and transported each week by the NMP to the national laboratory. The intended output was to obtain accurately barcoded filter papers for molecular analysis and linkage to individual case notifications. However, this again meant an additional workload for the healthcare staff, which might have a negative impact on their other activities. In the first quarter, poor sample collection, packaging and recording of the required patient details, large numbers of samples/ notification forms without barcodes and a low number of quality samples shipped to the central lab for molecular analysis were observed. This necessitated frequent refresher training, quality checks and supportive supervision. Despite the slow start, the overall outcomes of the number and quality of samples for molecular marking increased, for instance, the overall linkage of the molecular resistance markers and case notification rose from 51% in the first quarter to the highest of 75.1% in the third quarter, with participants noting how the activities strengthened the routine antimalarial resistance surveillance [12].

*'… remember we went with the supervisor to a couple of clinics when we identified that problem (positive mRDT sample with insufficient or missing DBS) and then we went back and told them that now we no longer just take the RDT. We also need to take the dry blood spots, so we need enough blood for the dry blood because sometimes we find that the dry blood is just a tiny bit of the blood and it really cannot be used. So we had to make sure that we go back to them and tell and train them that when you prick, make sure you prick enough just in case it is a positive you can also get more blood. They are currently a lot better'* **[Nurse, FGD03].**

**Table 1. Experience in collecting, labelling, packaging and shipping dried blood spots (DBS) as reported by 64 staff working in healthcare facilities in Nkomazi, Mpumalanga, South Africa.**

| | Not part of my work (%) | Easy for me (%) | Sometimes difficult (%) | Always difficult (%) |
|---|---|---|---|---|
| Labelling individual mRDTs | 7.8 | 43.8 | 39.1 | 9.4 |
| Collecting and labelling individual filter paper samples (blood spots) | 4.7 | 51.6 | 34.4 | 9.4 |
| Packaging positive and negative mRDTs for shipping | 4.7 | 65.6 | 17.2 | 12.5 |
| Packaging and shipping DBS samples? | 6.3 | 60.9 | 26.6 | 6.3 |
| Overall, how did you feel about the activities? | 5.9 | 55.1 | 29.5 | 9.4 |

As shown in Table 1, among the 64 survey participants who performed these tasks, the majority reported that overall, it is easy for them to collect, label, package and ship DBS samples (55.1%). However, 29.5% found it difficult sometimes, and 9.4% found it always difficult.

### 4.3 Process III: case investigation

Malaria case investigators aim to trace all confirmed malaria cases, obtain their residential GPS coordinates, identify malaria risk factors and actively screen nearby contacts for malaria. The additional intended output for the integration of molecular resistance markers into the routine malaria surveillance system was to enhance residential GPS coordinate data quality using *eTrex -10* GPS devices or Samsung tablets, with training and related SOPs [20]. Overall, 78.5% of malaria cases were successfully investigated (Fig 1), with the accuracy of GPS coordinates increasing from 48% in the first quarter to 89% in the last quarter [12]. The poor accuracy in the first two quarters was associated with device-related (non-uniform settings, device malfunctioning, insufficient battery charge), human-related (transcription errors, incomplete patient addresses) and system and resource-related factors (insufficient batteries or backup devices, increased workload, formatting lost during file transfers).

While discussing the issue of missing contact details, participants in FGDs mentioned that many of their patients are immigrants who lack phones, travel far to seek healthcare or do not know how to clearly describe the locations of their recently acquired residences. Additionally, due to their foreign and, at times, undocumented migrant status, many patients are often hesitant to share their contact details. These missing details lead to case investigation failure, resulting in more missing data.

The case investigators occasionally reach out to the notifying healthcare staff when faced with a case lacking contact information, aiming to gather additional details about the patient. This additional follow-up was received both positively and negatively. Positively, this would reduce the cost of case investigation since the NMP staff travel substantial distances for case investigation, and a few more details can assist in finding the index case's location more directly.

'We drive almost 40 km or more to find a case, then the phone number is wrong and there is only one line for the address. Just looking for one case, you take the whole day while other notifications are waiting.' [**Case investigator, FGD01**].

However, the practice of case investigators contacting clinic staff on their mobile phones was met negatively by some clinic staff, who perceived it as an invasion to their privacy, potentially threatening their working relationship. During a FGD a nurse stated:

'Yes, they do so [call] because in the notification book we also write our phone numbers when we notify a patient; if there is something missing they can call. Even if we are not on duty, yah, threatening our working relationship.' [**Nurse, FGD03**]

On the other hand, some clinic staff and case investigators reported that calling each other is not a problem, even after-hours calls, mentioning that it's useful and gives them a sense of duty. They have developed trust, and without phone calls, investigators would be travelling long distances to look for cases. One participant in an FGD with the nurses said:

*'…it is fine because they want information, and they want to know if we have referred the patient [to hospital] or if the patient is going home.'* **[Nurse, FGD03]**

Another challenge raised by the NMP staff was administrative inefficiencies resulting in multiple notification books at health facilities. These contribute to some tension. For instance, case investigators reported having collected the notification papers from the wrong notification book (outdated book/ not currently in use) at some larger facilities, where several notification books may be used simultaneously. Others failed to access the notification book during the facility visit due to it being misplaced. This is usually resolved by removing the outdated books and leaving the facility with only one notification book and informing the supervisors.

Participant feedback emphasised the need for regular refresher training on malaria surveillance, particularly focusing on malaria notification and accurate case information capture at the facility level. Common themes included the importance of providing clear guidance through updated SOPs and facility guidebooks, which participants noted were instrumental in improving their confidence and efficiency. While some participants highlighted the additional workload associated with learning the new SOPs, most acknowledged the benefits, reporting improved accuracy of coordinates and overall enhancements in the NMP's surveillance system functioning. These insights affirm the value of structured training and clear procedural support in addressing key gaps in malaria surveillance. An information officer in a FGD said:

*'The ownership thing you know when you understand exactly what it is you are capturing and how important it is, then it makes all the difference in terms of making sure it is on file, because how these things usually get presented … it is always just another study that we do not know how it is going to end up, who it is going to benefit. So, I think also that approach of you [the study team] is excellent as we know what this is … and what this is for. This is how important it is, it is ours only.* **[Information officer, FG02]**

## 4.4 Process IV: data capture

Malaria-related data are captured at the sub-district data capture centre daily. Malaria case investigators submit case report forms from both the healthcare facilities and their case investigation visits. The additional data input for the integration of molecular resistance markers into the routine malaria surveillance system was the capturing of the barcode number into the DHIS2 system. A barcode scanner was introduced to reduce workload and avoid transcription errors. Even after being trained, data clerks experienced challenges using the scanners, leading to inconsistencies in barcode capturing with both manual capturing and scanner use. The barcode scanning was also affected by internet connectivity, as any drop in WiFi signal disconnected the DHIS2 system. Overall, the process increased the workload and affected the perceived usefulness of the scanner, quoting a data clerk interviewed:

*'Typing is easier than using a barcode scanner'.* **[Data clerk, FGD01].**

Furthermore, delays in data capturing could last a few days to weeks. Delays were caused by interruptions in network connectivity, increased caseloads, and staff turnover:

*'When it is the rainy days and the cases increase, sometimes it's only two of us capturing data. Therefore, we would be late to capture… Late for a few days to one or two weeks.'* **[Data clerk, FGD01].**

In the first two quarters, there were significant missing details on the notification forms. For instance, not all forms had barcodes, coordinates or patients' addresses. Data clerks identified patterns of missing barcodes and helped identify whether facilities were underperforming or ran out of barcode stickers. This was communicated to the case investigation supervisors who provide supportive supervision to the healthcare workers notifying malaria. Supervisory staff reported that it remained common to find paper notification forms, mRDTs or DBS without barcode stickers due to forgetfulness, increased workload or reluctance from the clinic staff. Unidentifiable sample forms without barcodes were discarded as no information was provided for identification or linkage. Quoting one of the participants in a FGD:

*'Yes, that are missing, maybe the facility code or anything like that, then maybe three or four times we have experience that the RDT [referring to mRDT] came alone like just the RDT without the filter paper, without a barcode. So, most of the time we just discard because there is nothing we can do or go back and find this person and do the thing again'* **[Surveillance supervisor, FGD02]**

This resulted in low link-ability of samples because of inadequately barcoded samples, transcription errors and slow data-capturing processes. However, in the longer term, due to supportive supervision and re-training, cases linked at the household level increased with an increase in willingness and uptake of the programme [12].

### 4.5  Process V: analysis and reporting

The NMP collates and analyses malaria data at the provincial level on a monthly basis and shares the reports with the healthcare facility, district, provincial and national teams. For the integration of molecular resistance markers into the routine malaria surveillance system, the additional analyses included malaria case distribution and linkage of individual notifications to their molecular resistance markers (S3 Fig). The NMP information officers and molecular laboratory team downloaded data from the DHIS2 each month and shared it with the research team for further analysis and reporting 1) maps of the distribution of malaria cases; 2) linkage of residential locality and health facilities with molecular resistance markers and 3) data quality. If any gaps were identified, activities were formulated for further enhancing resistance surveillance.

A total of 4,787 malaria cases were notified and 78.5% (n = 3,758) of cases were investigated in the study period. However, data did not always match, and since these data came from separate files within the DHIS2 system, they needed to be merged outside the DHIS2 environment, leading to extra workload for information officers with a risk of non-matching, duplication, or deletions of some cases, variables or values. Commenting on the workload and how the DHIS2 framework could lead to non-matching of data and hence linkage failure, an NMP officer said:

*'…I need to merge the NMC form together with the case investigation form and to take it outside the computer and merge; even if it is Excel, it is prone to changing formats and that could introduce issues that could lead to non-matching within the DHIS2 framework.'* **[Information officer, KII03].**

S2 Fig compares data from DHIS2 and the two lower levels (health facility and sub-district data capture centre); the notified case counts were only consistent in 40% (6/15) of the notifications and 20% (3/15) for cases investigated. Most of the matching evaluations occurred in the final two study quarters. Looking at the discrepancies, the two secondary levels (reports at the data collection centre vs data in the DHIS2) had less data variability compared to their primary source (health facility). Although the study's quantitative analysis identified substantial duplication, the malaria team did data cleaning before sharing the datasets, as explained by one of the interviewees:

*'Once notifications were submitted and paper forms were captured in the DHIS2 system, ideally, all cases were merged and duplicates removed. In our analysis and feedback, some cases had duplicates that required further cleaning.'* **[Information officer, KII03].**

Analysis and reporting activities were part of routine practice before the integration of molecular resistance markers into the routine malaria surveillance system. However, due to the expansion of reported variables and the monthly downloads, merging and reporting required, this process became more complex and increased workload, especially in the first quarter. However, these extra activities were perceived as useful to the programme. The research team and the programme co-developed a training programme and data curation tools, engaging all involved in identifying gaps and improvements that further simplified the data analysis and reporting and improved the overall notification system [20]. This improvement reflected beyond the study and supported improvement for the programme reporting, quoting an IDI participant:

*'I think another thing that is good, our case investigation has also improved because I remember, when we started our baseline was 35% [for] 48 hours [time between case notification and investigation] and [for] 72 hours it was around 48%. So now on third quarter we reported above 65 (%) which was even above the target we set for this financial year. I think with them making the follow ups, having to go to the facilities to check on the stocks and everything has made them also be involved in going there and investigating the cases because the thing was XXXX and XXXX [SS4ME pilot programme staff names] would be analysing and supporting with feedback'* **[Supervisory staff, KII02]**

## 4. Discussion

This study identified factors affecting data quality, linkage, and consistency for integrating molecular antimalarial resistance monitoring into routine malaria surveillance in a pre-elimination malaria setting in sub-Saharan Africa. Healthcare facilities, district and provincial level healthcare staff in Nkomazi sub-district, Mpumalanga, South Africa were key beneficiaries, and their perspectives spearheaded this integration. Applying a process-oriented logic model, an iterative process for comprehensively analysing various factors that either facilitate or hinder successful integration was established, with end-to-end data flow and interaction with various users, beneficiaries, the existing systems and newly adopted technology. The programme's overall performance was influenced by cross-cutting factors such as workload, training, perception, supervision, leadership and resources – acting as enablers or barriers based on their availability, adequacy and perception.

As reported previously [12], 45% of malaria cases could not be linked to their molecular profiles or localities. Many of these individuals were migrants, often undocumented, who could not be followed up due to missing or inaccurate local addresses or phone numbers. Additional factors contributing to this non-linkage included errors in capturing location data, whether due to human mistakes or technical issues with devices.

Increased workload was identified as a cross-cutting factor for all processes. Adequately trained staff who perceived the activity as useful maintained high-performance levels, delivering better-quality outcomes even during months with a high malaria caseload, reduced staff and multiple reporting systems. In contrast, if staff did not perceive the activity as useful, data quality remained low, irrespective of resources and training provided. Previous studies have shown that a negative perception affects overall performance as well as the adoption and implementation of new technologies [21,22]. Low morale could lead to reluctance and other activities being prioritised, contributing to poor data quality.

The impact of devices (GPS devices, tablets, barcode scanners) introduced or updated for the integration of molecular resistance markers into the routine malaria surveillance system depended on the healthcare workers' skills and perception of the devices' usefulness and ease of use. Optimal functionality of devices coupled with adequate training enhanced performance, leading to observable improvements in data quality, especially during and immediately post-training and supervision periods [12]. Conversely, malfunctions in devices, such as internet disruptions, barcode scanner failure to connect to the DHIS2, depleted GPS device batteries, or clinic-level shortages of barcode stickers, significantly impacted team performance, morale, and data quality. Other studies exploring technological adoption in the health sector highlight

the crucial need to fully understand end-users' needs and potential useful functionalities to facilitate usage and enhance quality. These studies stress the importance of user-friendly functionalities and overall device performance [22,23].

This study found sustained supportive supervision and quality assurance tools and resources improved performance. These include adequate financing, long-term appointments without interruptions of contracts, a manageable workload and good collaboration among and between both government and non-governmental organisation staff. On the other hand, multiple parallel notification systems, communication challenges, inadequate supervision, lack of quality control tools, unmanageable workload, and poor collaboration negatively impacted intervention implementation.

The study highlighted challenges at different operational levels, specifically in the malaria notification and case investigation processes. Although the NMC Notification Book served as the gold standard for reporting malaria cases, the survey results revealed parallel and occasionally interchangeable use of notification systems. A 2019 review assessing the strengths and challenges of implementing DHIS2 across 11 Low- and Middle-Income Countries (LMICs) highlighted pervasive data quality issues, notably demonstrated in a Nigerian study where reported data in DHIS2 remained incomplete at least 40% of the time [24,25]. Despite encountering such data quality challenges in this study, more than half of the cases could be linked to resistance molecular marker data, with clear improvements in data quality observed over the study period [12].

Contrasting responses were observed regarding follow-up phone calls for additional information —some healthcare workers felt it assisted with case investigations while others felt it strained the working relationships, negatively impacting staff morale and performance. This contrast reflected varied staff perceptions and relationships. However, effective leadership, including the establishment of conducive communication channels, procedural guides, and efficient documentation systems, has successfully addressed similar challenges in comparable contexts [23]. This demonstrated the pivotal role of leadership and management in optimising communication and operations. Previous studies in similar settings have shown similar factors impacting the general adoption and implementation of information systems in health care [21,26–29]. A high level of perceived workload, data tools not being used as intended, and data quality issues were previously identified as challenges facing DHIS roll-out in Uganda and South Africa [30,31].

Our study acknowledges several limitations in diagnostic methods, evaluation processes, participant selection, and the analytical approach used for combining qualitative and quantitative data. As this study was embedded in the routine malaria surveillance system, diagnostics relied on the *hrp2-based* mRDT, which is *P. falciparum* specific. This may have led to non-*P. falciparum* malaria being missed. However, regular sample testing and species detection studies have shown *P. falciparum* to dominate in this area, where other species are rare [12]. Moreover, quality control measures, such as collecting 10% of negative mRDTs for further monitoring, including surveillance for hrp2/3 deletions and other malaria species, were implemented [12].

No single framework was comprehensive enough for the evaluation of the integration of molecular resistance markers into a routine malaria surveillance system. Incorporating diverse technologies such as barcode stickers, filter papers, GPS protocols, and scanners introduced complexity to the evaluation process. The 2021 United Kingdom Medical Research Council framework for developing and evaluating complex interventions offers a comprehensive evaluation spanning a study's lifecycle, from conception to implementation. However, it lacks detailed guidance on assessing individual components independently and primarily serves as a guiding framework without providing specific granularity for evaluation [32]. For the integration of molecular resistance markers into the routine malaria surveillance system, immediate end users are the healthcare staff and malaria case investigators. However, most determinant frameworks do not include end-users, and research has been scarce for improving the evaluation of how various end-users influence implementation effectiveness. Thus, our study employed an iterative process-oriented logic model.

Focusing on a process-oriented logic model can limit understanding of how individuals adapt to new interventions by emphasising operational steps over user perspectives. While effective for mapping interactions between technologies, people, and processes, it may overlook how healthcare workers and investigators interpret and adapt to molecular resistance surveillance in routine malaria notification within their unique sociocultural and institutional contexts. Therefore, this

approach may risk neglecting critical factors like trust, resistance, and informal workflows that emerge during implementation [33]. However, the iterative nature of this study and its focus on end-users allowed it to capture both functional factors (e.g., resistance) and relational factors (e.g., trust), addressing some of these limitations [34].

Our results may be biased as only 42 staff members from the participating healthcare facilities consented to participate in the FGDs and IDIs, while 64 participated in the survey. Most healthcare staff have multiple commitments, and their time is very limited. The study results might not fully reflect the view of all staff in Nkomazi sub-district and the generalisability of this study may be limited. However, the study team ensured diversity during participant recruitment to allow for different views and experiences and saturation was achieved with the diversity of perspectives, the depth of information provided, and the quality of dialogue. The latter, underpinned by a strong theoretical background, contributed to achieving high information power in the study. More studies might be needed to establish the extent to which the results from this setting are applicable in other pre-elimination contexts.

## 5. Conclusion and recommendation

Overall, the factors influencing the integration of molecular resistance markers into the routine malaria surveillance system in this sub-Saharan African pre-elimination setting were rooted in challenges inherent to the surveillance system itself, and were not specific to the integration of molecular resistance markers. This study underscores the intricate interplay of factors influencing malaria notification data quality and the integration of drug resistance markers within the routine surveillance system. It emphasizes the pivotal role of aligned perceptions, adequate resources, and supportive supervision in bolstering data quality, while also revealing the vulnerabilities stemming from device malfunctions, conflicting guidance, and disparate reporting systems. The evaluation frameworks used highlighted the need for more comprehensive models that consider the holistic healthcare environment, user perceptions, and nuanced interactions, particularly for complex interventions such as this. To enhance the implementation of such multifaceted interventions, future evaluations should include multiple countries to capture various healthcare settings, individual perceptions, and contextual nuances, enabling a more generalisable assessment and refinement of these interventions within routine healthcare systems.

## Supporting Information

**S1 Fig. Malaria Notification Systems: An illustration showing the different malaria notification systems involved during the study on integrating molecular markers of resistance into routine malaria notification system.** (DOCX)

**S2 Fig. Malaria Cases Notified and Investigated.** Comparison of malaria case notifications (S2a) and investigations (S2b) from source (Health Care Facilities – HCF), DCC and DHIS2 using three HCFs as the primary source. Malaria case data were aggregated and compared in the different levels for every first month of the five quarters in Nkomazi, Mpumalanga South Africa. (DOCX)

**S3 Fig. Linkage of individual patient data on malaria cases notified and their laboratory data: Data flow from notifiable medical condition (NMC) forms captured into the District Health Information System II (DHIS2), linked with their individual blood samples (malaria rapid diagnostic tests (mRDTs) and filter paper-dried blood spots) analysed using PCR for species confirmation and detection of molecular markers of antimalarial drug resistance.** (DOCX)

**S1 Tool. SS4ME Survey: A survey tool to collect primary healthcare staff's experience on additional activities introduced during SS4ME on integrating molecular markers of resistance into routine malaria notification system.** (DOCX)

**S2 Tool. Information sheet and consent forms for SS4ME Survey (S2a) and Semi-Structured Interviews for Focus Group Discussions (S2b) and In-depth Interviews (S2c).**
(DOCX)

**S3 Tool. Semi-structured guide for Focus Group Discussions (S3a) and In-depth Interviews (S3b) on integrating molecular surveillance for markers of resistance into the routine malaria notification system.**
(DOCX)

**S1 Table. A Process-oriented logic model for assessing the integration of early warning interventions in existing surveillance system.**
(DOCX)

## Acknowledgments

We would like to acknowledge the Mpumalanga Provincial and Nkomazi sub-district Malaria Control Programmes of the National Department of Health of South Africa, the Laboratory for Antimalarial Resistance Monitoring and Malaria Operational Research, Centre for Emerging Zoonotic and Parasitic Diseases, National Institute for Communicable Disease, Clinton Health Access Initiative, South Africa, Humana People to People (HPP), South Africa, and all NMP personnel and healthcare facility staff who collaborated in this study for all their support.

## Author contributions

**Conceptualization:** Frank M. Kagoro, Elizabeth Allen, Jaishree Raman, Richard J. Maude, Karen I Barnes.

**Data curation:** Frank M. Kagoro, Elizabeth Allen, Ray Magagula, Gerdalize Kok.

**Formal analysis:** Frank M. Kagoro, Jaishree Raman.

**Funding acquisition:** Philippe J. Guerin.

**Methodology:** Frank M. Kagoro, Jaishree Raman, Karen I Barnes.

**Project administration:** Aaron Mabuza.

**Validation:** Elizabeth Allen.

**Visualization:** Frank M. Kagoro, Richard J. Maude.

**Writing – original draft:** Frank M. Kagoro.

**Writing – review & editing:** Frank M. Kagoro, Elizabeth Allen, Jaishree Raman, Aaron Mabuza, Ray Magagula, Gerdalize Kok, Gillian Malatje, Philippe J. Guerin, Mehul Dhorda, Richard J. Maude, Karen I Barnes.

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
