## [Decision Letter · Decision Letter 0]

15 Aug 2024

PONE-D-24-18820Factors Affecting Integration of an Early Warning System for Antimalarial Drug Resistance within a Routine Surveillance System in a Pre-elimination Setting in Sub-Saharan Africa.PLOS ONE

Dear Dr. Barnes,

Thank you for submitting your manuscript to PLOS ONE. After careful consideration, we feel that it has merit but does not fully meet PLOS ONE’s publication criteria as it currently stands. Therefore, we invite you to submit a revised version of the manuscript that addresses the points raised during the review process.

We look forward to receiving your revised manuscript.

Kind regards,

Benedikt Ley, PhD

Academic Editor

PLOS ONE

3. We note that you have referenced (Barnes K, Mabuza A, Frean PJ, Magagula R, Bridget M, et al. Smart surveillance towards malaria elimination in Mpumalanga, South Africa (SS4ME): novel approaches for mapping antimalarial resistance (Protocol). 2017) which has currently not yet been accepted for publication. Please remove this from your References and amend this to state in the body of your manuscript: (ie “Bewick et al. [Unpublished]”) as detailed online in our guide for authors

Reviewers' comments:

Reviewer's Responses to Questions

**Comments to the Author**

1. Is the manuscript technically sound, and do the data support the conclusions?

Reviewer #1: Partly

Reviewer #2: Yes

Reviewer #3: Yes

Reviewer #4: Yes

2. Has the statistical analysis been performed appropriately and rigorously? 

Reviewer #1: N/A

Reviewer #2: Yes

Reviewer #3: N/A

Reviewer #4: Yes

3. Have the authors made all data underlying the findings in their manuscript fully available?

Reviewer #1: No

Reviewer #2: Yes

Reviewer #3: Yes

Reviewer #4: Yes

4. Is the manuscript presented in an intelligible fashion and written in standard English?

Reviewer #1: Yes

Reviewer #2: Yes

Reviewer #3: Yes

Reviewer #4: Yes

5. Review Comments to the Author

Reviewer #1: This is an interesting manuscript deserving publication in plos One. For the ease of the reader I have some suggestions to improve clarity:

The manuscript report data from the national malaria control program, however it seems that there are only 2 authors from the provincial program included. Affiliation 10 which seems to be a combination of the national (?) malaria program and the CHAI is not indicated.

The title suggest that this is a multi-country report across sub-Saharan Africa. It would be better for the authors to clearly identify in the title that this is a report from south Africa.

The same concern applied to the introduction (line 94-96) where it seems authors make a statement suggesting evidence derives from more than one location in South Africa.

Line 118-120 and remaining paragraph. Unclear how this framework looks like. No reference given. Better to provide some details for the reader to understand why it is limited. Also unclear here if you use the presented study to update and refine the framework.

Methods. It would be helpful for the reader to clearly differentiate between methods of the surveillance system and methods of the study presented.

Methods. Unclear based on what data the survey tool was developed. Was there any piloting done etc.

Methods: qual analysis (line 194) unclear what conceptual framework was used.

In the method section it seems that the three elements i) the monthly and quarterly analysis, ii) the survey and iii) the qual work are all presented. But then in the results it seems that i) is reported elsewhere already. Need to make clearer in methods already. Line 231 unclear if you are referring to the quarterly assessments mentioned under i) or to the surveys? Was the survey done multiple times?

Suggest to structure the results better. In line 224-227 you present some emerging themes from the qual data analysis and again in line 235-237 you present additional /different themes. Unclear if they came from different analysis. Its also unclear how you defined themes and key process a priori. Perhaps it would help the reader to have a clearer understanding of your process framework in the methods section and how this fits in with your analysis framework.

Discussion; a detailed discussion on how the approach of focusing on the process logic is a limitation to a more holistic understanding of how people make sense of new interventions /system would be beneficial.

Reviewer #2: Overall, the manuscript is well-written, engaging, and highly valuable to the malaria community. I have a few minor issues that need addressing or clarification, but I don’t have any major concerns.

Minor issues

Title:

Given that your study was conducted specifically in South Africa, it might be helpful to consider updating the section of title from "Pre-elimination Setting in Sub-Saharan Africa" to "Pre-elimination Setting in South Africa" for greater accuracy, provided it doesn’t complicate the process or other clearances

Result section

Clarity and Conciseness: the list of notification systems (lines 252-254) could be made clearer by rephrasing to emphasize their simultaneous use and impact. The sentence structure can be simplified to improve readability.

Impact of Additional Workload (Lines 271-275): While the increased burden is mentioned, provide more detail on how it affects surveillance quality, data accuracy, or patient care. This would link the workload to tangible outcomes or challenges

Details on Data Clerk Challenges (Lines 402-404): The issues with barcode scanning due to internet connectivity and the challenges faced even after training are important. Providing specific examples or further detail on these challenges could enhance understanding. For instance, how frequently were connectivity issues occurring, and what specific inconsistencies were noted?

Addressing Missing Details (Lines 416-423): Highlighting missing details and incomplete forms is important, but it's also essential to provide more information on the measures implemented to resolve these problems. For instance, what steps were taken to ensure that facilities received an adequate number of barcodes? Additionally, how were facilities informed about the significance of using barcodes?

Detailed Impact of Multiple Reporting Systems (Lines 526-533): The discussion of multiple reporting systems and their impact could be expanded. For instance, how did the presence of parallel systems affect data integration and overall reporting efficiency?

Reviewer #3: Editor in chief,

I had the pleasure of reviewing the manuscript entitled ‘’Factors Affecting Integration of an Early Warning System for Antimalarial Drug Resistance within a Routine Surveillance System in a Pre-elimination Setting in Sub- Saharan Africa’’ assigned manuscript number PONE-D-24-18820.

The manuscript is well articulated and comes at an appropriate time as many regions that were previously malaria-endemic are now moving toward pre-elimination phase. The manuscript highlights the importance of continued and sustained molecular surveillance as part of pre-elimination phase. It also identifies barriers to successful implementation and discussed potential challenges. However, I have several minor comments as indicated below.

Abstract

The abstract is well written and accurately links the research questions with the anticipated outcome. However,

1. The conclusion that reads ‘’ However, cross-cutting barriers inherent in the healthcare system can determine its success in a resource-limited setting’’. The message here is not clear. The author should consider using a different word like ‘’influence or affect’’, unless a different clarification of what the authors wish to convey to the reader is given.

Authors

2. Please use consistent formatting while writing authors with initials. For example, use John I. Smith instead of switching between ‘’John I Smith’’. The authors need to maintain consistency. Specifically, Frank M. Kagoro vs Richard J Maude. Please add a full stop after the initial.

Introduction

The introduction is well presented, indicating the existing gaps as well as the rationale for conducting the study.

Materials and Methods

The materials and methods section is well presented. However, there are a few areas that need improvement as indicated below:

3. The authors indicate that ‘’falciparum-specific histidine-rich protein 2 (HRP2)-based mRDT’’ was used. HRP2 mRDTs are only relevant to P. falciparum mono-infection, not mixed infections that need Pf/PAN mRDTs. The authors need to clarify why HRP2 mRDTs were preferred instead of combo mRDTs that would detect mixed infection or mono-infection of species other than P. falciparum. Species other than P. falciparum could also be circulating in the area and may be missed by the HRP2-only mRDTs. If the preference for HRP2 was due to the predominance of P. falciparum, this should be stated somewhere to justify this choice. Otherwise indicate this as one of the study limitations.

4. Furthermore, how was the issue of HRP2 deletion managed? Were there instances where some cases were mRDT- negative but slide or PCR positive? How were such situations handled? What was the gold standard of confirmatory detection tool that was used to ascertain mRDT results?

5. It is further indicated in the methods section that ‘’ For molecular surveillance, DBS filter papers were to be collected from all patients with positive mRDTs’’. How were false negative cases determined, particularly in situations where only HRP2 mRDT was used, potentially missing other infections, such as P. ovale, P. malariae, P. vivax and others? There is likelihood that a significant number of positive cases were missed.

6. Please also indicate the brand name and manufacturer of the mRDT used, as this is a standard practice when writing a manuscript.

7. In the main study, how was a malaria case defined? Using only mRDT seems inadequate. Were other tests such as malaria microscopy or species diagnostic PCR performed? If not what were the reasons for not doing so, and how did you differentiate mixed infections from P. falciparum mono-infection or infections other than P. falciparum given the fact that only HRP2 mRDT was used?

Results

8. Of the notified cases, 55.1% (n=2,636) were linked to their Plasmodium falciparum molecular marker resistance profiles, with 85% (n=2,240)…. Here around 45% of the cases could not be linked with the P. falciparum molecular marker profile. This could either be due to a large number of false positive cases by mRDT that did not generate any parasite DNA for genotyping or only 55% were purposively selected for PCR analysis. The author should clarify this by at least providing a CONSORT diagram indicating how the sampling was done while depicting samples for PCR analysis.

9. The authors should also indicate whether species diagnostic PCR was performed to rule out the presence of other species or mixed infection. Figure 1 is not very elaborate on this.

10. Lines 264-265 ‘’However, in responding to the survey question on which systems are used frequently for notification, 61/64 participants reported using these interchangeably: Malaria Connect (36, 59%), NMC Notification Book (15, 25%) and NMC Mobile Phone Application (3, 5%).’’

What is indicated in the parentheses needs to be specified, or this should be written as ‘’36 (59%)’’, as it can confuse the reader. In the text it is written as ‘’five (8%)’’ which could be presented as (five, 8%). Please consider revising this part to be more concise and clearer.

11. Line 426 ‘’ Yes, that are missing, maybe the facility code or anything like that, then maybe three or four times we have experience that the RDT [referring to mRDT] came alone like just the RDT without the’’. Please correct the tense! Use ‘’experienced’’

12. Line 436 ‘’ Routinely, the NMP collates and analysis malaria data at the provincial level each month and shares the’’. Please change the word analysis to ‘’analyses’’. Also, check the font size.

13. Figure 1: Please clarify the discrepancies observed between notified, investigated and linked cases. Specifically, explain the variability between 4787 vs 3758 and 2636. What were the methodological issues related to such variation? This needs to be well defined.

14. Figure 2: The visibility is blurry. Consider redrawing or reducing the stretching.

Reviewer #4: The manuscript is well-written and describes the research concisely. There are a few formatting issues. I have also suggested another way of addressing the stated bias with a comment that would speak to the qualitative research element of data saturation. Finally I am not convinced that having the participant's HCW category/role in the quotation adds value and may instead draw negative attention to the group.

6. PLOS authors have the option to publish the peer review history of their article (what does this mean? ). If published, this will include your full peer review and any attached files.

**Do you want your identity to be public for this peer review?** For information about this choice, including consent withdrawal, please see our Privacy Policy .

Reviewer #1: No

Reviewer #2: **Yes: ** Joseph Joachim Joseph

Reviewer #3: **Yes: ** DANIEL TR MINJA, NATIONAL INSTITUTE FOR MEDICAL RESEARCH, TANGA CENTRE, TAGA, TANZANIA

Reviewer #4: No

---

## [Author Response · Author response to Decision Letter 1]

30 Sep 2024

Dear Editor,

As requested, our manuscript has been revised in response to the reviewers' comments (also attached as separate document to include new figure S3).

Reviewer 1:

Abstract

The abstract is well written and accurately links the research questions with the

anticipated outcome. However,

1. The conclusion that reads ‘’However, cross-cutting barriers inherent in the

healthcare system can determine its success in a resource-limited setting’’ . The message here is not clear. The author should consider using a different word like ‘’influence or affect’’, unless a different clarification of what the authors wish to convey to the reader is given.

Response: We thank the reviewer for this suggestion, we have revised the paragraph (lines 73-74) to reflect this clarity.

“However, cross-cutting barriers inherent in the healthcare system can influence its success in a resource-limited setting.”

Authors

2. Please use consistent formatting while writing authors with initials. For example, use John I. Smith instead of switching between ‘’John I Smith’’. The authors need to maintain consistency. Specifically, Frank M. Kagoro vs Richard J Maude. Please add a full stop after the initial.

Response: The consistency has been noted. We have revised the authors’ names to reflect that on lines number 8-10.

Frank M. Kagoro 1,2,3,4,5, Elizabeth Allen 1,3,4, Jaishree Raman 6,7,8, Aaron Mabuza 1,3, Ray Magagula 9, Gerdalize Kok 9, Gillian Malatje 9, Philippe J. Guerin 3,4,5, Mehul Dhorda 2,3,4,5, Richard J. Maude 2,5,11,12, Karen I. Barnes 1,3,5

Materials and Methods

The materials and methods section is well presented. However, there are a few areas that need improvement as indicated below:

3. The authors indicate that ‘’falciparum-specific histidine-rich protein 2 (HRP2)-based mRDT’’ was used. HRP2 mRDTs are only relevant to P. falciparum mono-infection, not mixed infections that need Pf/PAN mRDTs. The authors need to clarify why HRP2 mRDTs were preferred instead of combo mRDTs that would detect mixed infection or mono-infection of species other than P. falciparum. Species other than P. falciparum could also be circulating in the area and may be missed by the HRP2-only mRDTs. If the preference for HRP2 was due to the predominance of P. falciparum, this should be stated somewhere to justify this choice. Otherwise indicate this as one of the study limitations.

Response: We thank the reviewers for noting this. This study used methods and data from the routine malaria notification system. As per the National Guideline and the Mpumalanga Malaria Programme, the test used during the study was HRP2-based, given the predominance of P. falciparum. The malaria programme also offers frequent training and quality assurance measures, such as collecting an additional 10% of the negative mRDTs that are sent to the NICD for polymerase chain reaction (PCR) for quality assurance. We have revised lines 143-147 for clarity as follows:

Individuals identified through proactive, reactive, or passive case detection were screened for malaria using a P. falciparum specific histidine-rich protein 2 (HRP2)-based mRDT (First Response™ Malaria Ag P. falciparum HRP2 Detection Rapid Card Test, Premier Medical Corporation Ltd, India) according to National Guidelines and the Mpumalanga Provincial tender process. An additional 10% of the negative mRDTs were collected and sent to the National Institute for Communicable Diseases (NICD) for quality assurance.

4. Furthermore, how was the issue of HRP2 deletion managed? Were there instances where some cases were mRDT- negative but slide or PCR positive? How were such situations handled? What was the gold standard of confirmatory detection tool that was used to ascertain mRDT results?

Response: During quality assurance testing, 10% of mRDT-negative samples were re-examined, and a small proportion (2.2%) were found to be PCR-positive. This was due to low parasite loads below the detection limit of mRDT (200 parasites/µl) but detectable by PCR (20 parasites/µl). Further investigations ruled out histidine-rich protein 2 (hrp2) deletions as the cause of false negatives. These findings are detailed in the quantitative paper [1], and the discussion has been updated to reflect this (lines 562-570):

As this study was embedded in the routine malaria notification system, diagnostics relied on the hrp2-based mRDT, which is P. falciparum specific. This may have led to non-P. falciparum malaria cases being missed. However, regular sample testing and species detection studies have shown P. falciparum to dominate in this area, where other species are rare. Moreover, quality control measures, such as collecting 10% of negative mRDTs for further monitoring, including surveillance for hrp2/3 deletions, detected only a small proportion (2.2%) of false negatives due to low parasite densities below the detection limit of mRDT (200 parasites/µl) but detectable by PCR (20 parasites/µl) [12]. Further investigations ruled out histidine-rich protein 2 (hrp2) deletions as the cause of false negative mRDTs.

5. It is further indicated in the methods section that ‘’For molecular surveillance, DBS filter papers were to be collected from all patients with positive mRDTs’’. How were false negative cases determined, particularly in situations where only HRP2 mRDT was used, potentially missing other infections, such as P. ovale, P. malariae, P. vivax and others? There is likelihood that a significant number of positive cases were missed.

Response: As responses to questions 3 and 4 mentioned, continuous evaluation of negative samples has shown other malaria species to be rare [1]. The malaria program also provided regular training and quality assurance, including collecting 10% of negative mRDT samples for PCR testing at the NICD. Results from these quality assurance measures are promptly integrated into national guidelines and reflected in practice. This information has been included in lines 143-147 (included in response to question 3 above).

6. Please also indicate the brand name and manufacturer of the mRDT used, as this is a standard practice when writing a manuscript.

Response: This has been specified (lines 144-145) as: “First Response™ Malaria Ag P. falciparum HRP2 Detection Rapid Card Test, Premier Medical Corporation Ltd, India”.

7. In the main study, how was a malaria case defined? Using only mRDT seems inadequate. Were other tests such as malaria microscopy or species diagnostic PCR performed? If not what were the reasons for not doing so, and how did you differentiate mixed infections from Pf mono-infection or infections other than Pf given the fact that only HRP2 mRDT was used?

Response: As noted above, routine diagnosis in primary healthcare facilities is based on HRP2 mRDT, molecular PCR analysis was used to confirm all malaria-positive and 10% of malaria negative infections. Of the 3748 malaria-positive RDTs, 13% (n = 477) were malaria-negative by PCR, five were pure Plasmodium malaria, and four were mixed infections (P. malariae and P. falciparum). This information has been reported previously as referenced [1] and described in the discussion as noted in the response to comment 4 above.

Results

8. Of the notified cases, 55.1% (n=2,636) were linked to their Plasmodium falciparum molecular marker resistance profiles, with 85% (n=2,240)…. Here around 45% of the cases could not be linked with the Pf molecular marker profile. This could either be due to a large number of false positive cases by mRDT that did not generate any parasite DNA for genotyping or only 55% were purposively selected for PCR analysis. The author should clarify this by at least providing a CONSORT diagram indicating how the sampling was done while depicting samples for PCR analysis.

Response: We have provided a new diagram below showing malaria cases and a sample flow chart and included in supplementary documents (Fig. S3), to explain this further, and have added a new paragraph in the discussion in lines 509-514:

As reported previously [1], 45% of malaria cases could not be linked to their molecular profiles or localities (Figure S3). Many of these individuals were migrants who could not be followed up by case investigators due to missing or inaccurate local addresses. Additional factors contributing to this non-linkage included errors in capturing residential location data, whether due to human mistakes or technical issues with devices, further complicating the reporting process.

Figure S3: Malaria cases detected and laboratory data linkage data flow from March 2018 to February 2020 in Nkomazi sub-district, Mpumalanga. The notifiable medical condition (NMC) forms captured the data on malaria cases and later fed into the District Health Information System II (DHIS2). Laboratory samples included malaria rapid diagnostic tests (mRDTs) with their respective filter paper-dried blood spots, which were further analysed using PCR for confirmation and species detection.

9. The authors should also indicate whether species diagnostic PCR was performed to rule out the presence of other species or mixed infection. Figure 1 is not very elaborate on this.

Response: Yes, PCR-based species diagnosis was carried out, as further explained in response to question 7 above and reported previously [1]. We have also added these details in Fig. S3 in the supporting information as explained above in response to comment 8.

10. Lines 264-265 ‘’However, in responding to the survey question on which systems are used frequently for notification, 61/64 participants reported using these interchangeably: Malaria Connect (36, 59%), NMC Notification Book (15, 25%) and NMC Mobile Phone Application (3, 5%).’’

What is indicated in the parentheses needs to be specified, or this should be written as ‘’36 (59%)’’, as it can confuse the reader. In the text it is written as ‘’five (8%)’’ which could be presented as (five, 8%). Please consider revising this part to be more concise and clearer.

Response: To avoid confusion, this description has been revised to capture that in lines 269-271.

However, in responding to the survey question on which systems are used frequently for notification, 61/64 participants reported using these interchangeably: Malaria Connect (36, 59%), NMC Notification Book (15, 25%) and NMC Mobile Phone Application (3, 5%) (Figure 3).

11. Line 426 ‘’Yes, that are missing, maybe the facility code or anything like that, then maybe three or four times we have experience that the RDT [referring to mRDT] came alone like just the RDT without the’’. Please correct the tense! Use ‘’experienced’’

Response: This quote captures verbatim the words used by the respondent, without grammatical correction.

12. Line 436 ‘’Routinely, the NMP collates and analysis malaria data at the provincial level each month and shares the’’. Please change the word analysis to ‘’analyses’’. Also, check the font size.

Response: This has been corrected as suggetsed.

14. Figure 2: The visibility is blurry. Consider redrawing or reducing the stretching

Response: Thank you for this recommendation. This figure has been revised as suggested.

Reviewer 2:

3. Materials and Methods

Line 138 1) instead of :)

Response: This typo has been corrected (line 137).

Line 141. Be consistent with how ‘sub-district’ is written

Response: All mentions of sub-district now have been revised to lower case except for the beginning of the sentence or a title.

Line 262: I'm not sure about using the HCW category as an identifier in the extracted quotes. Does it add value and will it in any way draw negative attention to that group?

Response: We would like to retain this grouping as it explains a major cadre responsible for malaria testing and notification at primary healthcare facilities. This cadre carries agency, role, and responsibility. Since they form most the workforce and respondents to this study, we do not expect any negative attribution linked to this explanation. Furthermore, the context of these quotes is conveyed when healthcare workers are distinguished from e.g. data capturers or surveillance officers.

Line 436: Typo ‘analysis’ instead of ‘analyzes’

Response: This has been corrected to ‘analyses’.

Line 447: Correct ‘wouldn’t’ to ‘would not’

Response: This typo has been corrected.

Best wishes

Karen & Frank

References:

1. Kagoro FM, Allen E, Mabuza A, Workman L, Magagula R, et al. Making data map-worthy—enhancing routine malaria data to support surveillance and mapping of Plasmodium falciparum anti-malarial resistance in a pre-elimination sub-Saharan African setting: a molecular and spatiotemporal epidemiology study. Malar J. 2022 Dec 1;21(1):1–19.

---

## [Decision Letter · Decision Letter 1]

4 Nov 2024

PONE-D-24-18820R1Factors Affecting Integration of an Early Warning System for Antimalarial Drug Resistance within a Routine Surveillance System in a Pre-elimination Setting in Sub-Saharan Africa.PLOS ONE

Dear Dr. Barnes,

Thank you for submitting your manuscript to PLOS ONE. After careful consideration, we feel that it has merit but does not fully meet PLOS ONE’s publication criteria as it currently stands. Therefore, we invite you to submit a revised version of the manuscript that addresses the points raised during the review process. Reviewer 1's comments do not appear to have been addressed. If this was intentional, please provide a justification. If it was an oversight, kindly revise the manuscript and include a detailed response addressing each reviewer's comments. Additionally, ensure that all previous responses are consolidated into one document.

We look forward to receiving your revised manuscript.

Kind regards,

Benedikt Ley, PhD

Academic Editor

PLOS ONE

Reviewers' comments:

Reviewer's Responses to Questions

**Comments to the Author**

1. If the authors have adequately addressed your comments raised in a previous round of review and you feel that this manuscript is now acceptable for publication, you may indicate that here to bypass the “Comments to the Author” section, enter your conflict of interest statement in the “Confidential to Editor” section, and submit your "Accept" recommendation.

Reviewer #1: (No Response)

Reviewer #2: (No Response)

Reviewer #4: All comments have been addressed

2. Is the manuscript technically sound, and do the data support the conclusions?

Reviewer #1: (No Response)

Reviewer #2: Yes

Reviewer #4: (No Response)

3. Has the statistical analysis been performed appropriately and rigorously? 

Reviewer #1: (No Response)

Reviewer #2: N/A

Reviewer #4: (No Response)

4. Have the authors made all data underlying the findings in their manuscript fully available?

Reviewer #1: (No Response)

Reviewer #2: Yes

Reviewer #4: (No Response)

5. Is the manuscript presented in an intelligible fashion and written in standard English?

Reviewer #1: (No Response)

Reviewer #2: Yes

Reviewer #4: (No Response)

6. Review Comments to the Author

Reviewer #1: My previous comments have not been addressed. Please kindly revise the manuscript based on the submitted comments to the previous version

Reviewer #2: Comments on the Manuscript

Title: Factors Affecting Integration of an Early Warning System for Antimalarial Drug Resistance within a Routine Surveillance System in a Pre-elimination Setting in Sub-Saharan Africa

The manuscript presented by Karen I Barnes et al. is well-written, engaging, and provides valuable insights to the malaria research community. Overall, I find the manuscript to be of high quality; however, I have a few minor issues that need addressing or clarification. There are no major concerns.

Minor Issues

Methods Section

1. Tool Development and Validation: Please provide a clearer description of how the tools (Tool S1, S2, etc.) were developed and validated for this specific study context. Including the criteria or methods used for validation would enhance understanding.

2. Tool References (Lines 164 and 171): When referencing tools (S1, S2, etc.), I recommend adding a brief summary after each reference (e.g., “staff survey tool”) to improve readability and context for the reader.

3. Timeframes (Line 170): Instead of “3-6 June 2020,” consider rephrasing to “between 3rd and 6th June 2020” for clarity and consistency in date formatting throughout the manuscript.

4. Quantitative Analysis (Lines 185 & 186): While R programming is specified for data analysis, it would be beneficial to briefly mention the statistical tests or models applied to detect spatiotemporal trends and assess consistency in the data.

Results Section

1. Case Notifications (Lines 211–215): The section concisely provides essential statistics; however, the reference to “Further quantitative results are reported elsewhere” may leave readers curious about missing details. It would be helpful to clarify the aspects not included in this manuscript.

2. Summary of Key Findings: Please consider briefly summarizing key findings from the "elsewhere" report to provide readers with a fuller understanding of what this study covers.

3. Participant Feedback on Training (Lines 388-400): I recommend summarizing participant feedback on training more clearly. Highlighting common themes or significant outliers could provide additional depth to this section.

Reviewer #4: (No Response)

7. PLOS authors have the option to publish the peer review history of their article (what does this mean? ). If published, this will include your full peer review and any attached files.

**Do you want your identity to be public for this peer review?** For information about this choice, including consent withdrawal, please see our Privacy Policy .

Reviewer #1: No

Reviewer #2: **Yes: ** Joseph Joachim Joseph

Reviewer #4: No

---

## [Author Response · Author response to Decision Letter 2]

6 Dec 2024

Comments on the Manuscript

Title: Factors Affecting Integration of an Early Warning System for Antimalarial Drug Resistance within a Routine Surveillance System in a Pre-elimination Setting in Sub-Saharan Africa

The manuscript presented by Karen I Barnes et al. is well-written, engaging, and provides valuable insights to the malaria research community. Overall, I find the manuscript to be of high quality; however, I have a few minor issues that need addressing or clarification. There are no major concerns.

Response: We thank the reviewer for acknowledging the quality of our manuscript and pointing to issues that would improve it’s quality. Below are our point by point response.

Minor Issues

Methods Section

1. Tool Development and Validation: Please provide a clearer description of how the tools (Tool S1, S2, etc.) were developed and validated for this specific study context. Including the criteria or methods used for validation would enhance understanding.

Answer: This content has now been added in the methodology section, Lines 164 – 169.

A paper-based survey tool (Tool S1) and consent form (Tool S2) were collaboratively developed by the study team and was revised by the Mpumalanga Provincial Malaria Elimination Programme. Tools S3 (FGD guide) and S4 (KII guide) were similarly developed, with input from subject matter experts to ensure relevance and appropriateness. All tools were piloted and field-validated in collaboration with the malaria programme team, who tested them, provided feedback, and ensured their alignment with the study objectives before formal deployment.

2. Tool References (Lines 164 and 171): When referencing tools (S1, S2, etc.), I recommend adding a brief summary after each reference (e.g., “staff survey tool”) to improve readability and context for the reader.

This has now been reflected in Lines 171 - 181.

‘Tool S1 (staff survey tool) was administered between 1st March and 30th June 2020 to primary healthcare facility staff treating malaria patients to evaluate their practice, perception, and experience of the integration of molecular resistance markers into routine malaria surveillance system activities. For FGDs and IDIs, various cadres of staff involved in malaria case management at healthcare facilities and in the NMP were invited to participate by email or phone call. At least one staff member performing any malaria-related activities was invited per healthcare facility. After obtaining verbal and written consent using Tools S2 (b) and (c), FGDs and IDIs were conducted in English from 3rd to 6th June 2020 using pre-prepared interview FGD and KII guides (Tools S3 and S4 respectively) to maintain consistency and quality. The audio recordings were securely stored on the password-controlled study computer and later transcribed. Two study investigators listened to audio recordings with reference to the transcribed scripts and resolved any interpretation conflicts or transcription errors by consensus.’

3. Timeframes (Line 170): Instead of “3-6 June 2020,” consider rephrasing to “between 3rd and 6th June 2020” for clarity and consistency in date formatting throughout the manuscript.

This has now been reflected and all dates in the manuscript use this format.

4. Quantitative Analysis (Lines 185 & 186): While R programming is specified for data analysis, it would be beneficial to briefly mention the statistical tests or models applied to detect spatiotemporal trends and assess consistency in the data.

We have revised this part to explain the type of analysis reported in a separate published article in Lines 192 - 195.

‘All quantitative data analyses were conducted using R programming language (versions 3.6 and 4.0). Further quantitative methods used include spatial temporal, trend and usability analyses that have been explained elsewhere.[12]’

For consistency the descriptive metrics, measurements and their justification are explicitly mentioned from lines 195 – 202.

‘For consistency, quarterly aggregates of notified cases and investigated cases from each healthcare facility sampled were compared at three levels (healthcare facility, sub-district data capture centre and provincial DHIS2 records) and the median difference was computed. Consistency was defined as an equal number of cases being reported at different levels, with a difference of +/- 5 cases allowed to account for delayed reporting. Since a consistency benchmark for integrating molecular resistance data into malaria case data had not been described before, this was established by adapting the internal consistency benchmark proposed by the WHO Data Quality Assurance guideline.[19] Data were then explored quantitatively and qualitatively to identify possible causes of inconsistency.’

Results Section

1. Case Notifications (Lines 211–215): The section concisely provides essential statistics; however, the reference to “Further quantitative results are reported elsewhere” may leave readers curious about missing details. It would be helpful to clarify the aspects not included in this manuscript.

Thank you for suggesting this improvement. This in now reflected in lines 221 – 222.

‘Further quantitative results of spatial temporal, trends, usability and molecular markers are reported elsewhere.[12]’

2. Summary of Key Findings: Please consider briefly summarizing key findings from the "elsewhere" report to provide readers with a fuller understanding of what this study covers.

Thank you. We have now included this summary from lines 221 to 227.

‘No validated or associated artemisinin-resistant Kelch-13 mutations were identified among the 2,385 PCR-positive samples, while nearly all 2,812 samples analysed for lumefantrine susceptibility carried the wild-type mdr1 86ASN and crt 76LYS alleles, potentially associated with reduced lumefantrine susceptibility. Concurrently, the spatial accuracy of malaria case coordinates improved, with the average nearest neighbour distance decreasing from 330 km in the second quarter to 35 km by the fifth quarter. Further quantitative results of spatial temporal, trends, usability and molecular markers are reported elsewhere.[12]’

3. Participant Feedback on Training (Lines 388-400): I recommend summarizing participant feedback on training more clearly. Highlighting common themes or significant outliers could provide additional depth to this section.

Response: Thank you for noting that, we have decided to reorganise this part and add depth with regards to training in lines 400 – 407

‘Participant feedback emphasised the need for regular refresher training on malaria surveillance, particularly focusing on malaria notification and accurate case information capture at the facility level. Common themes included the importance of providing clear guidance through updated SOPs and facility guidebooks, which participants noted were instrumental in improving their confidence and efficiency. While some participants highlighted the additional workload associated with learning the new SOPs, most acknowledged the benefits, reporting improved accuracy of coordinates and overall enhancements in the NMP’s surveillance system functioning. These insights affirm the value of structured training and clear procedural support in addressing key gaps in malaria surveillance.’

---

## [Decision Letter · Decision Letter 2]

13 Dec 2024

PONE-D-24-18820R2Factors Affecting Integration of an Early Warning System for Antimalarial Drug Resistance within a Routine Surveillance System in a Pre-elimination Setting in Sub-Saharan Africa.PLOS ONE

Dear Dr. Barnes,

Thank you for submitting your manuscript to PLOS ONE. After careful consideration, we feel that it has merit but does not fully meet PLOS ONE’s publication criteria as it currently stands. Therefore, we invite you to submit a revised version of the manuscript that addresses the points raised during the review process.

We look forward to receiving your revised manuscript.

Kind regards,

Benedikt Ley, PhD

Academic Editor

PLOS ONE

Additional Editor Comments :

Dear Author,

It seems as if the comments of Reviewer 1 have still not been addressed. I copy them below for your consideration. Please kindly address all comments of the reviewer in a detailed point-by-point reply before resubmitting. If you do not wish to address any of the comments made, kindly justify your decision.

Please address the following comments:

"I am not sure what happened here, but I reviewed the original version and provided comments. they werent adressed at all in the resubmission and I have flagged this. The reply to reviewer comments for the most recent version dont take my comments into account.

I copy the orginal comments here again. please note that line numbers refer to the original submission.

This is an interesting manuscript deserving publication in plos One. For the ease of the reader I have some suggestions to improve clarity:

The manuscript report data from the national malaria control program, however it seems that there are only 2 authors from the provincial program included. Affiliation 10 which seems to be a combination of the national (?) malaria program and the CHAI is not indicated.

The title suggest that this is a multi-country report across sub-Saharan Africa. It would be better for the authors to clearly identify in the title that this is a report from south Africa.

The same concern applied to the introduction (line 94-96) where it seems authors make a statement suggesting evidence derives from more than one location in South Africa.

Line 118-120 and remaining paragraph . Unclear how this framework looks like. No reference given. Better to provide some details for the reader to understand why it is limited. Also unclear here if you use the presented study to update and refine the framework.

Methods. It would be helpful for the reader to clearly differentiate between methods of the surveillance system and methods of the study presented.

Methods. Unclear based on what data the survey tool was developed. Was there any piloting done etc.

Methods: qual analysis (line 194) unclear what conceptual framework was used.

In the method section it seems that the three elements i) the monthly and quarterly analysis, ii) the survey and iii) the qual work are all presented. But then in the results it seems that i) is reported elsewhere already. Need to make clearer in methods already. Line 231 unclear if you are referring to the quarterly assessments mentioned under i) or to the surveys? Was the survey done multiple times?

Suggest to structure the results better. In line 224-227 you present some emerging themes from the qual data analysis and again in line 235-237 you present additional /different themes. Unclear if they came from different analysis. Its also unclear how you defined themes and key process a priori. Perhaps it would help the reader to have a clearer understanding of your process framework in the methods section and how this fits in with your analysis framework.

Discussion: a detailed discussion on how the approach of focusing on the process logic is a limitation to a more holistic understanding of how people make sense of new interventions /system would be beneficial."

Reviewers' comments:

Reviewer's Responses to Questions

**Comments to the Author**

1. If the authors have adequately addressed your comments raised in a previous round of review and you feel that this manuscript is now acceptable for publication, you may indicate that here to bypass the “Comments to the Author” section, enter your conflict of interest statement in the “Confidential to Editor” section, and submit your "Accept" recommendation.

Reviewer #1: (No Response)

2. Is the manuscript technically sound, and do the data support the conclusions?

Reviewer #1: Partly

3. Has the statistical analysis been performed appropriately and rigorously? 

Reviewer #1: N/A

4. Have the authors made all data underlying the findings in their manuscript fully available?

Reviewer #1: Yes

5. Is the manuscript presented in an intelligible fashion and written in standard English?

Reviewer #1: (No Response)

6. Review Comments to the Author

Reviewer #1: I am not sure what happened here, but I reviewed the original version and provided comments. they werent adressed at all in the resubmission and I have flagged this. The reply to reviewer comments for the most recent version dont take my comments into account.

I copy the orginal comments here again. please note that line numbers refer to the original submission.

This is an interesting manuscript deserving publication in plos One. For the ease of the reader I have some suggestions to improve clarity:

The manuscript report data from the national malaria control program, however it seems that there are only 2 authors from the provincial program included. Affiliation 10 which seems to be a combination of the national (?) malaria program and the CHAI is not indicated.

The title suggest that this is a multi-country report across sub-Saharan Africa. It would be better for the authors to clearly identify in the title that this is a report from south Africa.

The same concern applied to the introduction (line 94-96) where it seems authors make a statement suggesting evidence derives from more than one location in South Africa.

Line 118-120 and remaining paragraph . Unclear how this framework looks like. No reference given. Better to provide some details for the reader to understand why it is limited. Also unclear here if you use the presented study to update and refine the framework.

Methods. It would be helpful for the reader to clearly differentiate between methods of the surveillance system and methods of the study presented.

Methods. Unclear based on what data the survey tool was developed. Was there any piloting done etc.

Methods: qual analysis (line 194) unclear what conceptual framework was used.

In the method section it seems that the three elements i) the monthly and quarterly analysis, ii) the survey and iii) the qual work are all presented. But then in the results it seems that i) is reported elsewhere already. Need to make clearer in methods already. Line 231 unclear if you are referring to the quarterly assessments mentioned under i) or to the surveys? Was the survey done multiple times?

Suggest to structure the results better. In line 224-227 you present some emerging themes from the qual data analysis and again in line 235-237 you present additional /different themes. Unclear if they came from different analysis. Its also unclear how you defined themes and key process a priori. Perhaps it would help the reader to have a clearer understanding of your process framework in the methods section and how this fits in with your analysis framework.

Discussion; a detailed discussion on how the approach of focusing on the process logic is a limitation to a more holistic understanding of how people make sense of new interventions /system would be beneficial.

7. PLOS authors have the option to publish the peer review history of their article (what does this mean? ). If published, this will include your full peer review and any attached files.

**Do you want your identity to be public for this peer review?** For information about this choice, including consent withdrawal, please see our Privacy Policy .

Reviewer #1: No

---

## [Author Response · Author response to Decision Letter 3]

23 Jan 2025

Responses to Reviewer 1:

This is an interesting manuscript deserving publication in plos One. For the ease of the reader I have some suggestions to improve clarity:

Response: Thank you.

The manuscript report data from the national malaria control program, however it seems that there are only 2 authors from the provincial program included. Affiliation 10 which seems to be a combination of the national (?) malaria program and the CHAI is not indicated.

Response: We would like to point out, and as indicated in the affiliations, that co-authors 5, 6, and 7 work in the Mpumalanga Provincial Malaria Elimination Programme. Authors 2, 3, and 12 serve in the South African Malaria Elimination Committee supporting the National Department of Health.

The title suggest that this is a multi-country report across sub-Saharan Africa. It would be better for the authors to clearly identify in the title that this is a report from south Africa.

Response: The title clearly states this is a study in a pre-elimination setting in Sub-Saharan Africa and not across Sub-Saharan Africa. We would like to retain the title as it clearly states the context of the study and other contexts that would benefit from these results.

The same concern applied to the introduction (line 94-96) where it seems authors make a statement suggesting evidence derives from more than one location in South Africa.

Response: The second paragraph of the introduction unpacks the strategy for responding to antimalarial drug resistance, recommendations for integrating resistance monitoring in routine systems, and implications for pre-elimination settings. The following paragraph contextualises this information to the Nkomazi sub-district in Mpumalanga, our study area. Therefore, we find no reason for the readers to misinterpret.

Line 118-120 and remaining paragraph . Unclear how this framework looks like. No reference given. Better to provide some details for the reader to understand why it is limited. Also unclear here if you use the presented study to update and refine the framework.

Response: We have revised the manuscript from lines 119 – 122 to explain further about this framework.

For SS4ME, a conceptual framework was developed to explore and guide how the roll-out, adoption and utilisation of new and existing technologies would inform malaria elimination goals. The SS4ME rollout would involve optimising and linking existing technologies (molecular surveillance for markers of resistance) with key functions situated at clinics, such as malaria testing, treating, referral, and notification, and district / provincial level, namely collation of malaria notification metrics and case investigation.

On lines, 125 – 127, we have explained that this study wanted to revise and build further on this framework.

Therefore, this approach needed to be revised and expanded to identify internal, external and interactive factors that could affect the implementation and impact of the intervention.

Methods. It would be helpful for the reader to clearly differentiate between methods of the surveillance system and methods of the study presented.

Response: The introduction (lines 111 – 115) included surveillance as per the South African Malaria Guidelines. As clarified in lines 119-122 added in response to the comment, the innovation was in the linking of these routine activities.

As per national malaria treatment guidelines, all suspected malaria cases presenting to health facilities should be confirmed by a mRDT or microscopy (passive case detection) before treatment is administered.[17] Additionally, in pre-elimination areas of South Africa, the NMP routinely screens high-risk groups, such as migrant and mobile populations (proactive case detection), and households surrounding the residence of index cases (reactive case detection).[6,7]

The first paragraph of the materials and methods has summarised the methods of the study followed by subsequent paragraphs explaining components of surveillance and additional assessments involved.

Methods. Unclear based on what data the survey tool was developed. Was there any piloting done etc.

Response: All survey tools were collaboratively developed and tested in collaboration with the malaria programme. This information has been provided in lines 169 – 174.

A paper-based survey tool (Tool S1) was collaboratively developed by the study team and was revised by the Mpumalanga Malaria Programme and approved by the University of Cape Town Health Research Committee. Tools S3 (FGD guide) and S4 (KII guide) were similarly developed, with input from subject matter experts in the team to ensure relevance and appropriateness. All tools were piloted and field-validated in collaboration with the malaria programme team, who tested them, provided feedback, and ensured their alignment with the study objectives before formal deployment.

Methods: qual analysis (line 194) unclear what conceptual framework was used.

We have added further text (underlined) to make this more understandable in lines 207 – 210

For the qualitative data, each audio recording was transcribed and imported to NVivo 12 before being coded deductively, based on the conceptual framework (linking technologies, processes and key functions) and interview guides, and inductively from other observations. The first coding cycle assigned labels to text excerpts in the transcripts as lowest level ‘nodes’.

In the method section it seems that the three elements i) the monthly and quarterly analysis, ii) the survey and iii) the qual work are all presented. But then in the results it seems that i) is reported elsewhere already. Need to make clearer in methods already. Line 231 unclear if you are referring to the quarterly assessments mentioned under i) or to the surveys? Was the survey done multiple times?

Response. All the monthly and quarterly assessments and survey and qualitative results are presented in this manuscript. Only spatial temporal analysis, trends and molecular markers are reported in the previously published manuscript. This is captured on lines 232 – 233.

Further quantitative results of spatial temporal, trends, usability and molecular markers are reported elsewhere.[12]

Suggest to structure the results better. In line 224-227 you present some emerging themes from the qual data analysis and again in line 235-237 you present additional /different themes. Unclear if they came from different analysis. Its also unclear how you defined themes and key process a priori. Perhaps it would help the reader to have a clearer understanding of your process framework in the methods section and how this fits in with your analysis framework.

Response: We have corrected a typo to clarify the key processes identified (lines 247-250).

Several key processes emerged from discussions on the integration of molecular resistance markers into the routine malaria surveillance system, which included malaria case notification, sample collection, packaging and transportation, case investigation, data capture and reporting.

The themes and subthemes are explained, lines 258 – 261.

In Fig. 2 below, prevalent themes in the FGDs and IDIs included ease of use, perceived usefulness of surveillance, staff members’ reluctance to adopt new activities, contradictions in best practice definitions, workload, and system support. These themes converged into sub-themes: work commitment, agency and ownership, challenging processes, compromise, staffing needs, training, leadership, and supervision.

Discussion: a detailed discussion on how the approach of focusing on the process logic is a limitation to a more holistic understanding of how people make sense of new interventions /system would be beneficial."

Response: We have added a paragraph to discuss this limitation, lines 603 - 610 .

Focusing on a process-oriented logic model can limit understanding of how individuals adapt to new interventions by emphasising operational steps over user perspectives. While effective for mapping interactions between technologies, people, and processes, it may overlook how healthcare workers and investigators interpret and adapt to molecular resistance markers in routine malaria notification within their unique sociocultural and institutional contexts. Therefore, this approach may risk neglecting critical factors like trust, resistance, and informal workflows that emerge during implementation. However, the iterative nature of this study and its focus on end-users allowed it to capture both functional factors (e.g., resistance) and relational factors (e.g., trust), addressing some of these limitations.

---

## [Decision Letter · Decision Letter 3]

31 Jan 2025

PONE-D-24-18820R3Factors Affecting Integration of an Early Warning System for Antimalarial Drug Resistance within a Routine Surveillance System in a Pre-elimination Setting in Sub-Saharan Africa.PLOS ONE

Dear Dr. Barnes,

Thank you for submitting your manuscript to PLOS ONE. After careful consideration, we feel that it has merit but does not fully meet PLOS ONE’s publication criteria as it currently stands. Therefore, we invite you to submit a revised version of the manuscript that addresses the points raised during the review process.

Kindly address all of teh provided comments of reviwer 1.

We look forward to receiving your revised manuscript.

Kind regards,

Benedikt Ley, PhD

Academic Editor

PLOS ONE

Journal Requirements:

Reviewers' comments:

Reviewer's Responses to Questions

**Comments to the Author**

1. If the authors have adequately addressed your comments raised in a previous round of review and you feel that this manuscript is now acceptable for publication, you may indicate that here to bypass the “Comments to the Author” section, enter your conflict of interest statement in the “Confidential to Editor” section, and submit your "Accept" recommendation.

Reviewer #1: (No Response)

2. Is the manuscript technically sound, and do the data support the conclusions?

Reviewer #1: Partly

3. Has the statistical analysis been performed appropriately and rigorously? 

Reviewer #1: N/A

4. Have the authors made all data underlying the findings in their manuscript fully available?

Reviewer #1: Yes

5. Is the manuscript presented in an intelligible fashion and written in standard English?

Reviewer #1: Yes

6. Review Comments to the Author

Reviewer #1: Thank you very much for giving me the opportunity to review the paper again. I am aware that this manuscript has gone through multiple iterations already, but I feel that some of my comments have still not been addressed adequately.

My concerns are particularly regarding the description and use of the conceptual framework. While the authors state now in the introduction (why not methods?) that they developed a conceptual framework designed to explore and guide the roll out, adoption and utilization of technologies would inform malaria elimination goals, it is unclear how this framework looks like. Perhaps the authors could add this framework as a figure to the supplement? The authors later state that the developed framework lacked options to explore interaction of determinants of success and how they interact with the users and state that the approach needed to be revised, but it remains unclear how the framework was changed. They then talk about process-oriented logic models, but its not clear at this stage that such a model was used (again, a figure showing the model might be helpful for the reader).

In the methods the authors than state that they used the conceptual framework to guide the qual data analyses. Its unclear which framework they refer to at this stage, e.g. the initially developed framework, the revised one or process-oriented logic model… Given the lack of clarity around the used framework (the authors added some key words in brackets that hint at the elements of the framework) it remains difficult for the reader to understand the analysis approach. In the discussion the authors refer again to an iterative process-oriented logic model, but this doesn’t align with the terminology used previously.

I appreciate the added section around the limitations of the framework (which is here called again iterative process-oriented logic model). However, this section would benefit from some references.

7. PLOS authors have the option to publish the peer review history of their article (what does this mean? ). If published, this will include your full peer review and any attached files.

**Do you want your identity to be public for this peer review?** For information about this choice, including consent withdrawal, please see our Privacy Policy .

Reviewer #1: No

---

## [Author Response · Author response to Decision Letter 4]

6 Mar 2025

Responses to Reviewer 1:

Thank you for your detailed review. We appreciate your feedback and the opportunity to clarify our use of the conceptual framework and process-oriented logic model used.

1. Clarification of the conceptual framework (SS4ME) and process-oriented logic model (Molecular Surveillance) used:

The implementation for the SS4ME study was designed to explore the roll-out, adoption, and utilization of technologies for malaria elimination. It focused on high-level determinants: problem identification, treatment-related issues, community-based interventions, outcomes, and impact. However, this implementation did not specifically account for the integration of molecular resistance markers into routine surveillance—the focus of this sub-study. To better evaluate barriers and enablers, an iterative process-oriented logic model was developed to track the roll-out of these components and assess their interactions with the surveillance system.

2. Justification for Placing the Conceptual Framework (SS4ME implementation) in the Background rather than Methods Section

The SS4ME implementation was discussed in the background to provide context for the sub-study reported in this manuscript. In order to avoid confusion and for clarity, we have revised the background from lines 117 – 119 and 123-125 to avoid reference to that conceptual framework, as stated below:

o For SS4ME, implementation focused on exploring and guiding how the roll-out, adoption and utilisation of new and existing technologies would advance malaria elimination goals.

o As with other implementation research, this roll-out lacked a mechanism to explore the interaction between the different determinants of success and the users or beneficiaries.

This sub-study focuses on the feasibility of linking molecular resistance markers within the malaria surveillance system, requiring a different framework tailored to assessing barriers, enablers, and implementation challenges. The methodology section has been revised to make this distinction and describe the process-oriented logic model used for this sub-study, as shown further below.

3. Reviewer's Request for a Figure

Structured summary tables of the process-oriented logic model have been included as Tables S1 – S2 to visually represent the key evaluated components and their interactions. We have revised the text in lines 207 - 209 to remove the confusion introduced by mention of the SS4ME conceptual framework and explicitly reference this figure in the methodology section.

o For the qualitative data, each audio recording was transcribed and imported to NVivo 12 before being coded deductively, based on the interview guides and the process-oriented logic model (Tables S1), and inductively from other observations.

The process-oriented logic model was developed to systematically analyse:

Inputs (e.g., resources, infrastructure),

Activities (e.g., implementation steps, training),

Outputs (e.g., data linkages, user practices),

Outcomes (e.g., integration success, surveillance improvements).

This allowed for a more structured evaluation of implementation barriers and enablers.

We have revised to include these examples for further clarity from lines 222-228:

o As a complex intervention, the study used a process-oriented logic model to link the overall qualitative and quantitative data to determine the enablers and barriers that would explain the changing trends. To do so, results were further categorised into processes and logically analysed to identify inputs (e.g., resources, technology components) and activities (e.g., implementation steps, training), intended and unintended outputs (e.g., data linkage, use practices), and outcomes (e.g., integration success, surveillance improvements). A matrix was constructed to explore the level of influence and performance for each key feature identified from the process logic model and is further presented in the results section.

4. References for the discussion

We have revised this section (lines 612-619) with references added for further clarity:

o Focusing on a process-oriented logic model can limit understanding of how individuals adapt to new interventions by emphasising operational steps over user perspectives. While effective for mapping interactions between technologies, people, and processes, it may overlook how healthcare workers and investigators interpret and adapt to molecular resistance markers in routine malaria notification within their unique sociocultural and institutional contexts. Therefore, this approach may risk neglecting critical factors like trust, resistance, and informal workflows that emerge during implementation.[33] However, the iterative nature of this study and its focus on end-users allowed it to capture both functional factors (e.g., resistance) and relational factors (e.g., trust), addressing some of these limitations.[34]

We have ensured all the clarifications made are in the final manuscript. Thank you for your constructive feedback.

---

## [Decision Letter · Decision Letter 4]

12 Mar 2025

Factors Affecting Integration of an Early Warning System for Antimalarial Drug Resistance within a Routine Surveillance System in a Pre-elimination Setting in Sub-Saharan Africa.

PONE-D-24-18820R4

Dear Dr. Barnes,

We’re pleased to inform you that your manuscript has been judged scientifically suitable for publication and will be formally accepted for publication once it meets all outstanding technical requirements.

Kind regards,

Benedikt Ley, PhD

Academic Editor

PLOS ONE

Additional Editor Comments (optional):

Reviewers' comments:

Reviewer's Responses to Questions

**Comments to the Author**

1. If the authors have adequately addressed your comments raised in a previous round of review and you feel that this manuscript is now acceptable for publication, you may indicate that here to bypass the “Comments to the Author” section, enter your conflict of interest statement in the “Confidential to Editor” section, and submit your "Accept" recommendation.

Reviewer #1: All comments have been addressed

2. Is the manuscript technically sound, and do the data support the conclusions?

Reviewer #1: Yes

3. Has the statistical analysis been performed appropriately and rigorously? 

Reviewer #1: N/A

4. Have the authors made all data underlying the findings in their manuscript fully available?

Reviewer #1: No

5. Is the manuscript presented in an intelligible fashion and written in standard English?

Reviewer #1: Yes

6. Review Comments to the Author

Reviewer #1: (No Response)

7. PLOS authors have the option to publish the peer review history of their article (what does this mean? ). If published, this will include your full peer review and any attached files.

**Do you want your identity to be public for this peer review?** For information about this choice, including consent withdrawal, please see our Privacy Policy .

Reviewer #1: No

---

## [Editor Report · Acceptance letter]

PONE-D-24-18820R4

PLOS ONE

Dear Dr. Barnes,

I'm pleased to inform you that your manuscript has been deemed suitable for publication in PLOS ONE. Congratulations! Your manuscript is now being handed over to our production team.

Kind regards,

on behalf of

Dr Benedikt Ley

Academic Editor

PLOS ONE